# A Framework for Service Semantic Description Based on Knowledge Graph

**Qitong Sun, Jun Han * and Dianfu Ma**

School of Computer Science and Engineering, Beihang University, Beijing 100191, China;
sunqt@buaa.edu.cn (Q.S.); dfma@buaa.edu.cn (D.M.)
* Correspondence: jun_han@buaa.edu.cn

**Abstract:** To construct a large-scale service knowledge graph is necessary. We propose a method, namely semantic information extension, for service knowledge graphs. We insist on the information of services described by Web Services Description Language (WSDL) and we design the ontology layer of web service knowledge graph and construct the service graph, and using the WSDL document data set, the generated service knowledge graph contains 3738 service entities. In particular, our method can give a full performance to its effect in service discovery. To evaluate our approach, we conducted two sets of experiments to explore the relationship between services and classify services that develop by service descriptions. We constructed two experimental data sets, then designed and trained two different deep neural networks for the two tasks to extract the semantics of the natural language used in the service discovery task. In the prediction task of exploring the relationship between services, the prediction accuracy rate reached 95.1%, and in the service classification experiment, the accuracy rate of TOP5 reached 60.8%. Our experience shows that the service knowledge graph has additional advantages over traditional file storage when managing additional semantic information is effective and the new service representation method is helpful for service discovery and composition tasks.

**Keywords:** web service; knowledge graph; ontology; deep neural network

## 1. Introduction

With the emergence of web services, developers can focus more on a core function and hand over other functions to others. It makes full use of network resources, reduces a lot of repetitive labor, and improves productivity and portability while reducing costs.

Service providers use WSDL to write documents, describe the interface information of the web services, and register the web service in the service registry. The service registry acts as a centralized collection of a large number of service information and publishes them to the customers of Web services. In this mode, there are two main ways for people to find the services that they need. One is to select the required services under some categories by searching the registry based on service name and classification. The second way is to search keywords directly in a search engine, and then seek the web services needed among massive websites.

In fact, more information means that people could choose more fine-grained services. In the process of service discovery, the information extracted by the registry is little, which leads to fuzzy retrieval conditions. Some web service information is directly presented to the consumers who need to carry out the service discovery task by manually reading the WSDL document. The lack of a proper way to organize different information regarding service also aggravates the information dispersion and isolation. In the case of no further effective condition, we can only choose the most suitable web service by letting professionals read the WSDL documents one by one.

In the absence of service information as a condition for further filtering, there is no intuitive centralized service information display, which greatly reduces the efficiency of service discovery and makes service composition difficult. Because WSDL provides a description of the service information, it cannot describe the relationship between services. However, the associative retrieval method provided by the information of relation is of great significance for service discovery and composition.

The emergence of large-scale knowledge graphs provides an opportunity to solve these problems. The application of knowledge graphs in various fields solves the problem of data scattered in multiple systems in the field. At the same time, the inherent advantages of graphs for the management of relationships between entities are more conducive to the use of relationships for correlation and association, which greatly improves the quality of the search. To resolve the problems, namely the lack of centralized web service information, the low efficiency of service discovery caused by no fixed search mode, and the high threshold of service discovery caused by domain-specific language, it is imperative to construct a large service knowledge graph.

We design and construct a service knowledge graph in this paper. Our main contributions are as follows: firstly, we design the service ontology of the service knowledge graph to represent service information. On the designed service ontology, we used the WSDL file data set provided by Zhang et al. to construct a knowledge graph containing 3738 service entities [1]. After that, we proposed a method of extending semantic information on knowledge graph. We conducted two experiments according to this method, constructed two classification data sets, and implemented a relationship discovery model and a label generator based on natural language description. Finally, we draw a conclusion that method of service discovery and service composition based on knowledge graph.

The rest of this paper is organized as follows: The second chapter summarizes the work related to service representation and deep learning. The design of service ontology and the construction of service knowledge graph are described in detail in the third chapter. The fourth chapter describes the semantic relationship extension method on the knowledge graph, and two experiments based on this method. We draw a conclusion in the fifth chapter.

## 2. Related Work

Web service description has gone through three stages: keyword based, syntax based and semantic based. The methods of the first two stages are inefficient and inaccurate. Semantic and analysis of services can improve the description and composition efficiency of services. In order to solve the above problems, researchers have done a lot of research on the discovery, description and matching of semantic information and services from different aspects. The research on service composition and recommendation has made a lot of achievements. A matching method based on the matching level between concepts proposed by Paolucci et al. [2]. This method mainly describes the semantic of Web services in DAML-S, so that the rich matching relationship between input and output concepts of services can be divided into four levels, namely Exact, Plug-in, Subsume and Fail. K. Pal. describes the ontology based semantic web service architecture and establishes a prototype system [3]. Description logic is used to represent the knowledge of terms in a structured way. The system is composed of structured case-based reasoning, rule-based reasoning and service concept matching algorithm. N. Archint et al. proposed a graph based semantic web service composition system [4], including two subsystems: time management and runtime. The time management subsystem is responsible for the preparation of dependency graph, in which the dependency graph of related services is automatically generated according to the proposed semantic matching rules. The runtime subsystem is responsible for using graph-based search algorithms to discover potential web services and non-redundant web service combinations that users query. Abid et al. proposed a web service composition framework based on semantic similarity calculation [5]. A service matching engine is established. Firstly, the engine classifies web services into class with similar functions by

using the similarity measurement. Then, through the composite service sequence matching with the target, the similarity computing technology is established in the two tasks. S. A. Khanam et al. proposed a new semantic web service discovery scheme [6], which uses the WSDL specification and ontology to determine the similarity between query and service, and uses the improved Hungarian algorithm to quickly find the maximum. The proposed method adopts the data structure, operation structure, and natural language description for information retrieval.

When extending the semantic relationship of service knowledge graph, we use deep learning technology to embed the service profile described in natural language, and output the tag of service. The feature of the recurrent neural network (RNN) is that it is very good at processing sequence data and can effectively integrate the text context information [7]. It combines the feedforward neural network and the feedback network path. The feedforward neural network is a simple network forward transmission process, and the input of each time step is the output hidden state of the previous time step. The input and output come from the same network structure, so as to effectively retain the known text information; while the feedback neural network is a kind of gradient descent algorithm (BPTT algorithm) which propagates along the time step, that is, the gradient value is backpropagation along the time. However, RNN has the problem of long-term dependence, that is, it cannot capture the information of a long time ago at the current time step, so it cannot effectively use the previous information. Some scholars have proposed many evolutionary structures with new RNN cells, including long short-term memory (LSTM) [8], gated recurrent unit (GRU) [9], etc. The idea of this kind of solution is to select the inflow and outflow of information at each time step through the gate. In the above discussion, we only discussed the one-way information transmission process, that is, the unidirectional transmission process from the beginning to the end of the text. However, the information that unidirectional flow can get is not enough. Schuster and Paliwal brought bidirectional network into people's sight and proposed a bidirectional recurrent neural network (BRNN) [10]. The network build in this structure can fully integrate the above and following information of the text to obtain the most reasonable prediction. Graves and Schmidhuber combined LSTM and BRNN to obtain bidirectional LSTM (BiLSTM) [11].

Transformer structure abandons the traditional structure of convolutional neural network (CNN) [12,13] and RNN. It obtains the weight of value by calculating the similarity between query and key. It completely depends on the design of attention, so that it can obtain the information of all words in a text at the same time, which improves the training efficiency [14]. Compared with RNN, it is better at extracting long sentences and paragraphs. Based on the structure of transformer, Jacob Devlin proposed a new language model called Bert, which sets standards for bidirectional encoder representations from transformers [15]. The language model is pre-trained from unlabeled text. For different downstream tasks, Bert can be combined with different output layers to build a new network model, and then fine tune the whole network by the downstream task data. The training cost of downstream tasks is greatly reduced, and some tasks with insufficient resources can also benefit from it. Bert achieved the best results at that time in 11 natural language tasks. Lan et al. improved the Bert model and proposed two parameter reduction methods to reduce memory consumption, and improved the training speed of Bert, which made the model more extensible, and SQuAD benchmarks while having fewer parameters compared with BERT-large [16].

Service representation is the premise of service discovery. In order to provide more meaningful results of service discovery, we propose a brand-new framework. We express service information by constructing service ontologies and store the service information on a large-scale knowledge graph. The user's input during retrieval is processed by neural networks.

Guodong et al. [17] also constructed a knowledge graph for service discovery. The service ontology in the knowledge graph contains five types of information. They identify service entities from structured and semi-structured data and extract service information. The information about the service is stored on the knowledge graph, and then the mapping template is designed according to the structure of the ontology. By designing a matching template, the service discovery mechanism is transformed into a query on knowledge graph. They identify the service entities of user's input, and use the categories of these entities as the search keyword, which reduces the threshold of service discovery to some certain extent, but the design of the ontology limits content used in search. Huang et al. [18] proposed an effective approach that combines the embeddings from language models (ELMo) representation and convolutional neural network (CNN) to obtain a more accurate similarity score for retrieving target web services. More specifically, first, they adopt the ELMo model to generate effective word representations for capturing the sufficient information from services and queries. Then, the word representations are used to compose a similarity matrix, which will be taken as the input for the CNN to learn the matching relationships. Finally, the combination of the ELMo representation and CNN is used to address the representation and interaction processes within the matching task to improve the service discovery performance. Lu [19] proposed a novel web service discovery method based on semantic similarity measure combing I/O similarity and context similarity. A new similarity measurement between terms is put forward as the basis of Web service similarity. The similarity between terms is represented by the similarity of two concepts containing them, and this is calculated by means of the length of the shortest path between two concepts.

Shen et al. [20] proposed a service discovery approach based on web service clustering. Web service discovery was divided into two steps: web service clustering and web service selection. In the process of web service clustering, the descriptions of web services were represented as vectors. In order to make the vector carry as much as possible, the semantic information, they tried four different unsupervised sentence representations. In another part, Latent Dirichlet Allocation (LDA) was used to mine topic semantic information of web services after user's web request was placed into a specific cluster according to its vector. The final efficiency of web service discovery was used to measure the effectiveness of their approach.

## 3. WSDL Syntax Ontology

### 3.1. Message and Datatype

When the service is published, the service is described as a series of endpoints for message operation through the WSDL file. Developers can get the information of the interface by reading the WSDL file published by the service provider, so as to realize the call. The WSDL file is in XML format. The three main elements in the WSDL file are type, operation, and binding. In addition, it includes definition, datatype, message, port type, port, service, document, import. Compared with WSDL, our whole service knowledge graph should have similar ability to describe the basic information of service. In the service knowledge graph, we design service ontology, message ontology, method ontology, datatype ontology and other ontologies. We use the relationship between ontologies to describe different services. In the service knowledge graph, we set the message ontology to represent the message element in the WSDL file abstractly. The message element in the WSDL document is used to describe the data exchanged between the service provider and the service consumer. Each web service contains two types of messages. One is the parameter of web service and the other is the return value of web service. There are part elements in the message element, and each part contains type attribute. The values of this attribute can be divided into three categories: one is the data type defined by the service through the type element, which is unique in the current service's namespace; the second is the data type defined in other documents and import by the current document; the last one is the basic data type in the XSD (XML schema definition) namespace. Under the type

node of WSDL document, XML schema is used to define new element types. There are two types of new element types, one is complexType, the other is element. ComplexType contains several elements and attributes. Element cannot contain other elements, but can only contain one certain type of text information, which can be XSD built-in type or a user-defined pattern. In the knowledge graph we designed, we did not adopt this kind of classification. The unified use of datatype ontology to describe complexType and element does not affect our complete description of service information. The datatype ontology contains the type attribute, and the values are the six most common built-in data types in XSD: string, decimal, integer, boolean, date, time. In addition to the type attribute, there is a complex attribute, which is a boolean type.

When the attribute value of complex of a datatype entity is false, the datatype is equivalent to element in the WSDL document, and its type attribute is one of the traditional data types. When the value of the complex is true, the entity has the same meaning as the complex node defined in the WSDL document. Taking this node as the source node, it establishes inclusion relationship with several datatype entities, and can establish inherit relationship with another node whose complex is true. There is no difference in the meaning between the abstract data types created in this way and the data types created by type. Moreover, there is no difference between the data types defined in other documents introduced through the namespace and those defined by the type node directly. Therefore, the description ability of the graph in the type definition is not weaker than that of the WSDL. The datatype entity created in the graph will be referenced by message, and the part relationship between message and datatype is set. We represent the different messages defined in the document as different message entities in the service knowledge graph, and each message entity has name attribute. We take the datatype entity as the source node of the part relationship, and the message which refers to the datatype entity as the target node. When we define the message, datatype, and the relationship between them, the knowledge graph has ability to store the various types of data that web services need to use. The structure of ontology is shown in Figure 1.

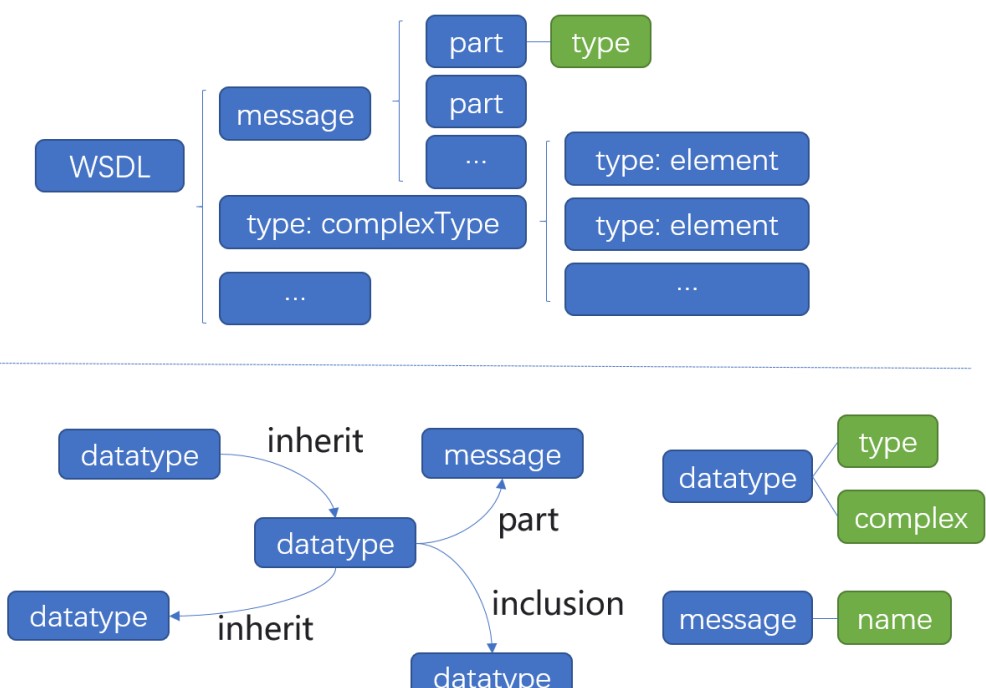

**Figure 1.** The message and type elements expressed in Web Services Description Language (WSDL) are located at the top. The corresponding message, datatype and relationship defined between them in service knowledge graph are located below. The green box indicates that the content is an attribute, and the blue indicates element (in WSDL) or ontology (in service knowledge graph).

The data definition described by WSDL is difficult to reuse to some extent, because service writers often don't retrieve whether there are defined classes with some component elements that are just needed. At the same time, because web services are described by WSDL documents, even if each service is described by only one WSDL file (while in reality, there may be several WSDL documents for one service), web service information is still too discrete, because the information between various services is difficult to coordinate, and the information is scattered in several files. Therefore, it is necessary to analyze the WSDL document of each service, and then summarize some aspects of information. This not only makes it difficult for custom to find services and classify services, but also makes it difficult for the registration authority of web services to provide service information query on a certain aspect. The way of using WSDL document is not conducive to the integration of a large number of web service information, so it is not conducive to retrieve and discover the defined data classes in different documents. Different service providers of the same type of service may define a large number of data with the same meaning and composition, resulting in redundancy. Only different services of the same service provider or different versions of the same service may have certain reuse, which leads to low utilization rate of data types defined by different web services. When the service knowledge graph is constructed, this phenomenon is improved. Each web service described by a WSDL document is represented as a node in the graph, which can integrate a large number of different web services. The graph of integrated information is conducive to the retrieval and discovery of message nodes with some characteristics. When the message node that meets the condition is found, the input relationship can be established directly with the service node to achieve node reuse. At the same time, in the graph, after the relationship between different services and the same data type node is established, it is more conducive to intuitively discover the additional semantic relationship between services, and expand the semantics of the graph. For example, Paolucci et al. divide the relationship between services into four types by comparing the input and output of different services [2]. This comparison is reflected in the graph that different services refer to the same message entity.

### 3.2. Method

Next, we set the method ontology in the graph to represent the operations element. In a WSDL document, operations are a child node of porttype and is used to describe the function name in the service interface, the message used as input, the message used as output, and the message encapsulated when an error occurs. Consumers know the information of callable interface methods provided by web services by parsing information of operation. In the graph, we do not set the porttype ontology alone, but directly create the method ontology of the service. Several operations defined in the WSDL document of a service are embodied as several method entities in the knowledge graph. We define the input relationship, output relationship and fault relationship between method and message. Among them, the source node of the input is a message, target node is the method with this message entity as input, and output or fault takes a method as the source node and the message output by this method as the target node. In a WSDL document, there are four basic patterns of operation: one-way, request response, solicit response, and notification. When describing operation, these four patterns reflect whether the operation has input or output sub nodes and the order of the sub nodes. In the service knowledge graph, the pattern attribute is set in the method ontology, and the attribute value is used to record this information. The contain relationship is established between service and method. The contain relationship is directed from service to method. A service entity can establish a contain relationship with several method entities. The structure of ontology is shown in Figure 2.

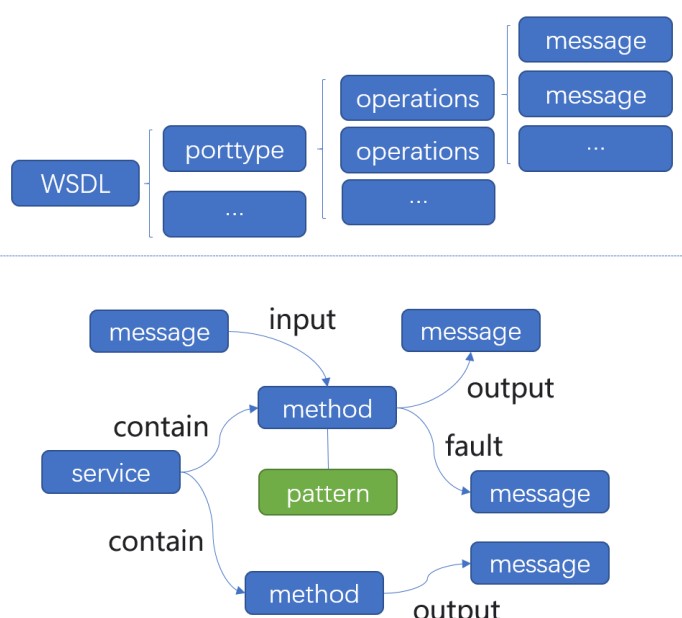

**Figure 2.** The porttype and operation elements expressed in WSDL are located at the top. The method and relationship defined between method and other ontology in service knowledge graph are shown below. The green box indicates that the content is an attribute, and the blue indicates element (in WSDL) or ontology (in service knowledge graph).

### 3.3. Binding

After defining the datatype, message and method, the binding ontology is set up in the graph to express the information about how the data in the WSDL document is transmitted in the network. The binding element in the WSDL document fully describe the protocol, serialization and coding. Different service providers of the same type of service can customize the binding to be distinguished from each other. Zero, one or more binding elements can be contained in one WSDL. The service element in WSDL is composed of a set of port elements. Each port element is associated with an address. At the same time, each port element will refer to a binding element. If the same binding element is referenced by multiple ports, it means that an additional URL address can be used as a replacement. In the service knowledge graph, bind relationship is established between binding ontology and method ontology. A binding entity can establish bind relationship with multiple method entities. The binding entity acts as the target node, and the method entity acts as the source node. Each binding entity has address attribute, which is used to specify the URL of the port, and contains the transport attribute to specify the protocol used in the transmission. Each bind relationship contains soapaction and style attributes. Soapaction indicates the action parameter carried in the application/soap + xml Content-Type header field. Style indicates the default style of this particular Simple Object Access Protocol (SOAP) operation.

In WSDL, there are header, body, fault, headerfault elements defined in the SOAP protocol namespace under the binding element. Therefore, in the service knowledge graph, we set header, body, fault, and headerfault relationships between the binding ontology and the message ontology. Each relationship has use and encodingStyle attributes. Use attribute is used for indicates how message parts are encoded. EncodingStyle indicates a particular encoding style to use. The structure of ontology is shown in Figure 3.

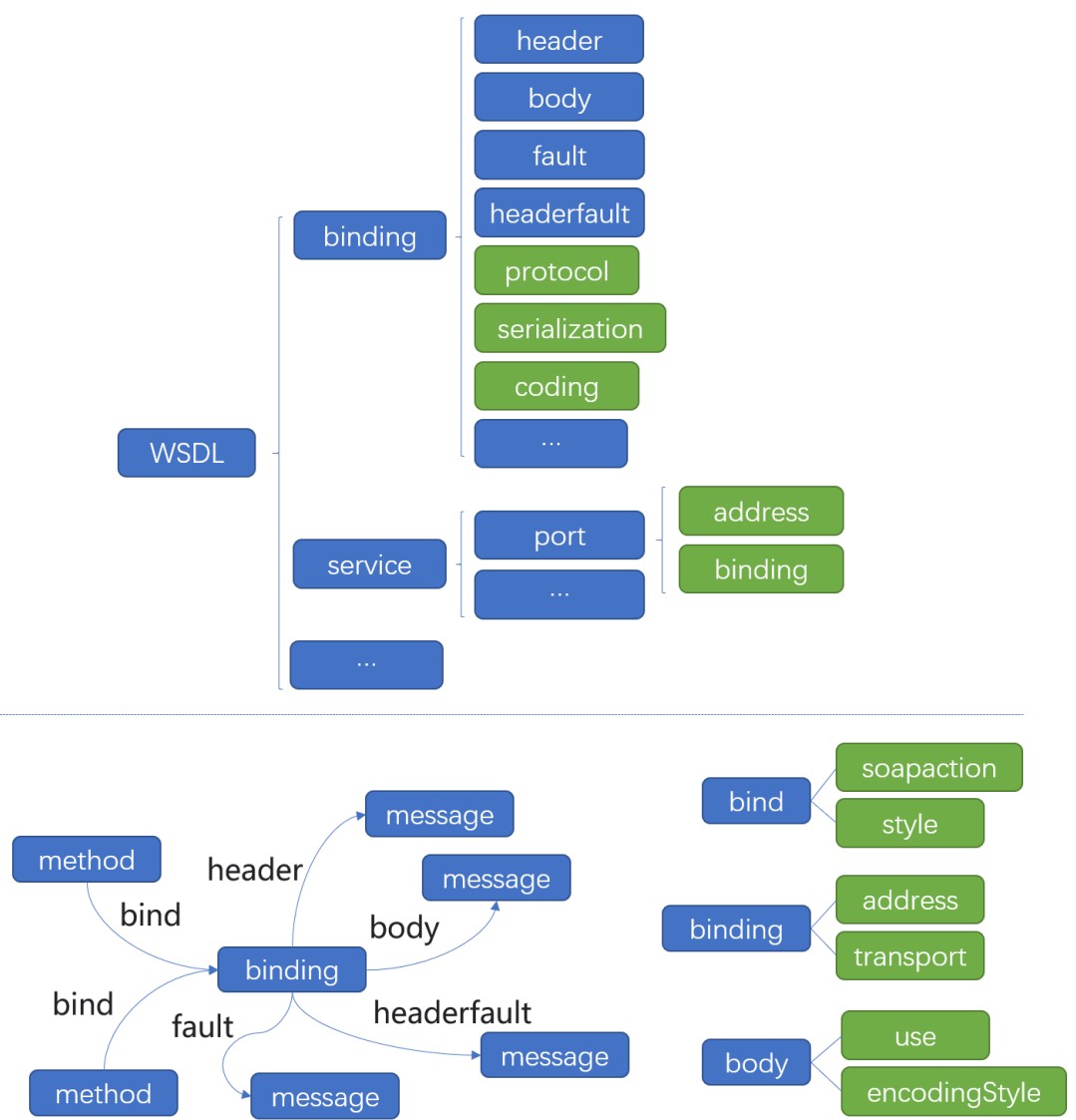

**Figure 3.** The binding and service elements expressed in WSDL are located at the top. The binding and relationship defined between binding and other ontology in service knowledge graph are shown below. The green box indicates that the content is an attribute, and the blue indicates element (in WSDL) or ontology (in service knowledge graph).

Finally, definition, the root element in the WSDL document is reflected as one whole service node in the graph, and the web service described by a WSDL document is embodied as a service entity in the knowledge graph. The service knowledge graph can integrate the information in a large number of web services' WSDL documents.

We have noticed that the key point of using a knowledge graph to store service information is that we should make full use of the characteristics of the format of the information that can be stored in the knowledge graph, and reasonable arrangement and design of ontology, attributes, and relationships in the graph can ensure that there will be no difference in the amount of information when the two methods are used to store and display the service information. In other words, when using the knowledge graph to represent the web service, there will be no information that can be covered in the WSDL file, but the knowledge graph cannot be expressed or stored.

*3.4. Ontology Instantiation*

In this chapter, we describe the service ontologies that are necessary to construct a large-scale service knowledge graph. While introducing these service ontologies, we de-

scribe the service information stored in each of these structures, and correspond to the way of describing information using WSDL documents.

When constructing a large-scale service knowledge graph based on the ontologies, we use the WSDL document data set provided by Zhang et al. [1] and the information of WSDL document is stored by the instantiation of the corresponding service ontology. We built a translation module to convert the WSDL document into information described by RDF, and store the information in the format of Turtle. When some information in the WSDL is empty, the instantiation of the information in the graph does not exist or the value of the attribute storing the information is empty. It depends on the form of information expression. As the information that does not originally belong to the same namespace will eventually be aggregated, in the construction of the knowledge graph, each service ontology has id and namespace attributes in addition to the relationships and attributes introduced earlier to ensure that there will be no conflicts in namespace when a large number of WSDL documents are automatically converted. The system finally converted the WSDL documents, and constructed a knowledge graph that contains 3738 service entities. An example of instantiation of a method ontology and the corresponding part of the WSDL document are shown in Figure 4. The construction method of the knowledge graph is shown in Figure 5.

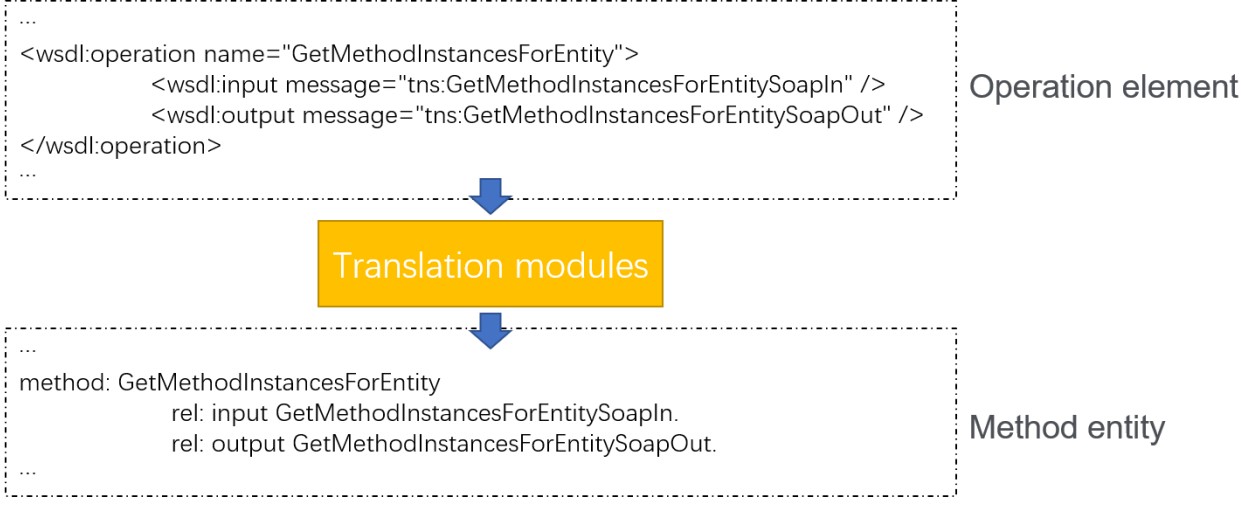

**Figure 4.** An example of converting the operation information stored in WSDL into a method entity.

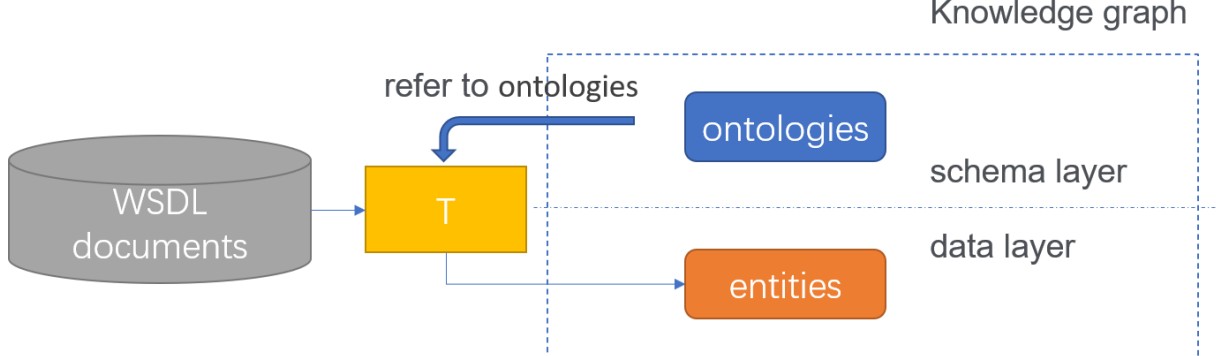

**Figure 5.** The method of constructing knowledge graph. T represents a series of translation modules. Designed ontologies form the schema layer of the knowledge graph. The translation module designed with reference to the ontologies converts WSDL documents into service entities and stores them in the knowledge graph.

## 4. Semantic Relation Extension on Knowledge Graph

### 4.1. Methodology

In the previous section, we can draw a conclusion that we can use a knowledge graph to store all kinds of web service information without omission, and sort out the service information more intuitively in the knowledge graph by defining ontology and relationship. However, the advantages of creating a service knowledge graph are not only that but also more convenient to store additional information by using a knowledge graph than using a WSDL file. We can store additional information in the service knowledge graph, which cannot be stored in the WSDL document. This information contains semantics, such as the relationship between service entities, the category of the service, and so on. Integrate this kind of information on the knowledge graph and use this information in downstream tasks. This process is called the semantic relationship expansion of the service knowledge graph. The process is shown in the Figure 6. The process of semantic relationship expansion is divided into two parts, building process and discovery process. In the building process, we need to determine the semantic information that need to be integrated into the knowledge graph and clarify the manifestation of this kind of information in the knowledge graph, and then store the information of each service entity. After that, we use various corpora on the network to label the data set about the semantics and train the neural network. In the discovery process, we input the natural language used in the search into the neural network, and the network obtains the output. Then use the query language conversion module set according to the semantic information to convert the network output into the corresponding graph query language, search in the knowledge graph, and finally realize the service discovery on the service knowledge graph using natural language. Under the guidance of this methodology, we conducted two experiments to expand the semantics of the service knowledge graph in two aspects.

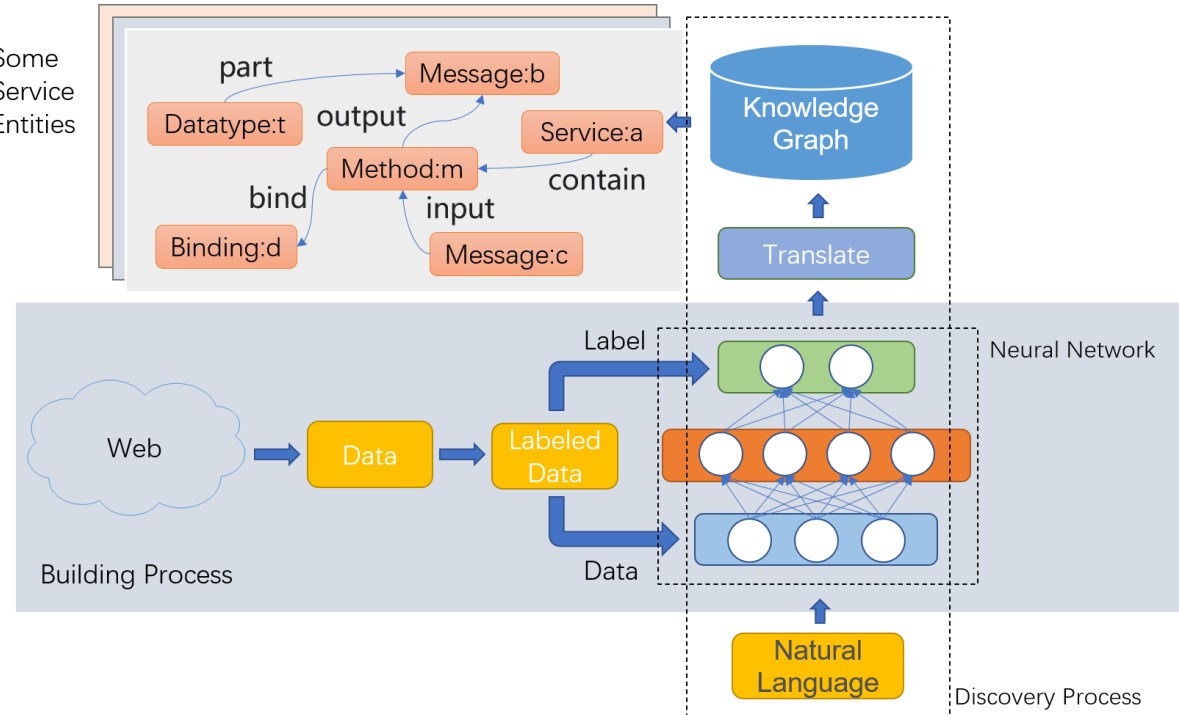

**Figure 6.** Process of semantic relationship expansion of the service knowledge graph. The whole process is divided into two parts: building and discovery. Build is the process used when expanding the semantics. The main task is to train the neural network. The discovery process is the application of newly added semantic information in the graph when doing service discovery tasks.

## 4.2. Semantic Relationship Detection

### 4.2.1. Dataset

Judgment of the semantic relationship between services is particularly critical in searching. More accurate judgments can better discover related web services and play an important role in service discovery and service composition tasks.

In the semantic relationship detection experiment, we try to use a neural network to automatically determine whether two services have an association relationship. The judgment is based on the introduction and description of the two services. We obtained data from the Internet and performed manual labeling, and constructed a data set containing 1,116,994 pieces of data. Examples of data are shown in Table 1. Intro1 and Intro2 mean the introduction from two services respectively. When the value of a tag is 1, it indicates that there is a semantic relationship between the two services, and there is no relationship when the value is zero. After the data are divided, the ratio among the training set, development set, and test set is 18:1:1.

**Table 1.** Sample data in the Semantic Relationship Detection experiment.

| Intro1 | Intro2 | Tag |
|---|---|---|
| Let Vimeo take care of all your video needs. Their API handles uploading, transcoding, hosting and playback of any video type. Embed the videos in your own website, or share them with the large community of filmmakers that call Vimeo home. The API uses RESTful calls and responses are formatted in JSON. | The Wikia Intelligent Search Extensions API is a JavaScript API that allows third-party sites to build applications that appear as Wikia Search results for end-users who install them. | 0 |
| The Google Custom Search Engine API is a RESTful API that allows developer to get web or image search results data in JSON or Atom format. With the API, developers can add web search and/or site search capabilities to their website, blog or collection of websites. | 3X Software Limited is a multi-purpose internet and retail services provider. The Postcodes 4 U API, provided by 3xsoftware, enables users to search the Royal Mail PAF database for an address from a postcode. The API uses REST calls and returns JSON or XML. The API supports calls to return an address from a postal code. | 1 |

### 4.2.2. Neural Network Structure

In the field of deep learning, more data means better results. Transfer learning can make fuller use of data resources. Since the network near the input in the model tries to extract the syntactic features of the text, the syntactic features of the training data of the same language do not change when finetune the network for specific downstream tasks. BERT, which uses a large amount of text for pre-training, has fully learned syntax and grammar, which can effectively reduce the training burden of downstream tasks, and achieve high accuracy in a short time, which is a good choice. Our network model adopts the method of combining the BERT and BiLSTM, using BERT's fitting of language grammar knowledge and BiLSTM's fitting of sentence semantics to judge whether there is a semantic relationship between the two services. The network structure is shown in Figure 7.

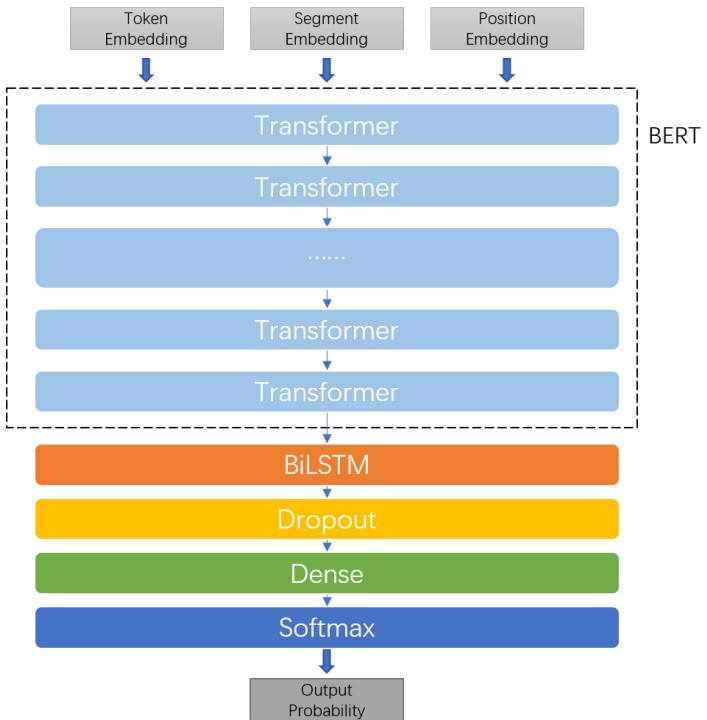

**Figure 7.** Classification network structure in the Semantic Relationship Detection experiment.

The input of the network is the embedding of natural language profiles from two services in three dimensions, namely token embedding, segment embedding and position embedding. Among them, segment embedding is used to distinguish two sentences, token embedding is used to represent the embedding of each token in the sentence, and position embedding is used to record the position information of the current token in the sentence. The input first enters the BERT, which is a series of 12 transformer layers, and the output of BERT enters the bidirectional LSTM, trying to extract the semantic relationship between the two introductions. After one layer of dropout and two layers of fully connected layers, the output is finally obtained from softmax. The output is a judgment on the semantic relationship.

4.2.3. Results

The training process is shown in Figure 8. The final accuracy rate of the model on the test set is 95.1%. The type of the task in this experiment belongs to the sentence pair classification task. The best accuracy of each traditional sentence pair classification task is shown in Table 2. It shows that the accuracy of the network has reached state of the art.

**Table 2.** Tasks and corresponding evaluation indicators.

| Task | Leaderboard |
|---|---|
| Natural Language Inference on MNLI | 92% (MATCHED) |
| Question Answering on QQP | 92.3% (ACCURACY) |
| Natural Language Inference on QNLI | 99.2% (ACCURACY) |
| Semantic Textual Similarity on STS-B | 92.9% (PEARSON CORRELATION) |
| Semantic Textual Similarity on MRPC | 93.7% (ACCURACY) |
| Natural Language Inference on RTE | 93.2% (ACCURACY) |
| Semantic Relationship between Services | 95.1% (ACCURACY) |

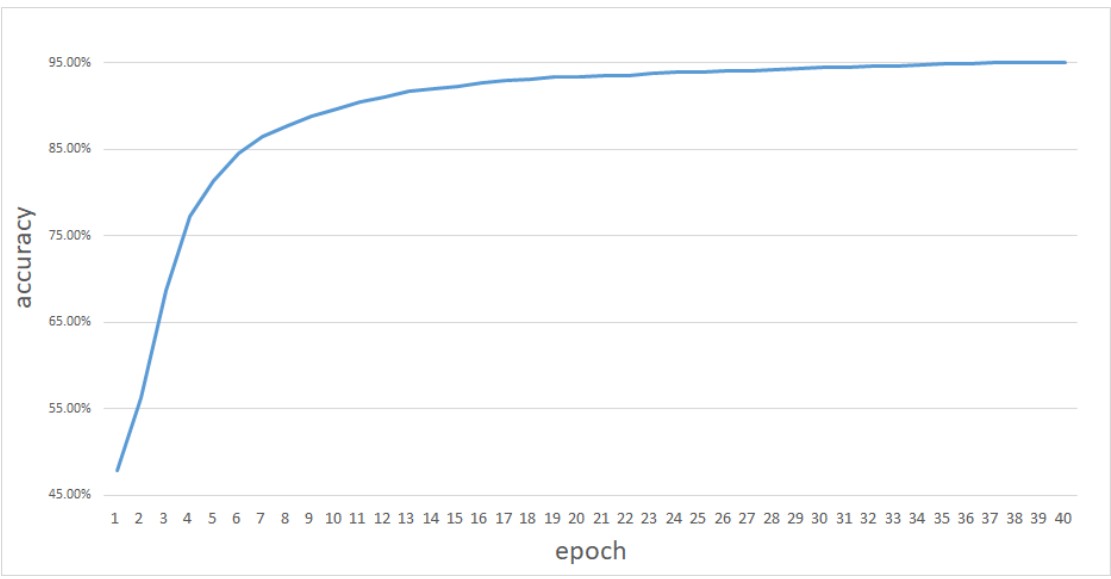

**Figure 8.** The accuracy changes with epochs.

It can be observed from Figure 8 that the pre-training effect of the network model has been demonstrated in the early stage of the finetune stage, and the accuracy has improved rapidly. At the same time, because there are only two types of labels in this experiment, the mathematical expectation of accuracy at the beginning of training is 50%. The actual initial accuracy is 47.92%. The final accuracy rate does not reach 100% because the current network scale cannot fitting all natural language description scenarios. The network still has insufficient semantic understanding. We get a better accuracy in this experiment compared with many other traditional sentence pair classification tasks, such as semantic textual similarity (STS). The reason for achieving better accuracy in this experiment is the difference between single sentence and paragraph. The data containing one or two single sentences in the traditional task are less semantic. For example, in semantic textual similarity on STS-B task, the predicted value is the judgment of the semantic relationship between two single sentences. In the experiment with a semantic relationship between services, each profile is composed of several sentences, and the network can obtain richer semantic information from them, so it has better accuracy compare with many traditional tasks. After getting the network prediction, we pass the output to the translate module and convert it into a graph query language. The accuracy of network prediction is the accuracy of the semantic content search at the time of service discovery.

*4.3. Service Classification*

4.3.1. Dataset

The classification of services is a very important information. The classification tag briefly summarizes the functions of services and is of great significance for the discovery and composition of services. People can select the desired services according to the classification labels as soon as possible. Therefore, we extend the semantics of service classification on the knowledge graph. We attach tags to each service entity node, and we use deep learning technology to build a tag generator, which generates tags by inputting a description of the service and uses tags for retrieval.

In the service classification experiment, we constructed a data set containing 72,320 pieces of data and 483 categories. Each piece is composed of the service introduction and the corresponding classification label. If a service corresponds to multiple categories at the same time, each category corresponds to one piece of data. The data structure is shown in Table 3. After the data are divided, the ratio among the training set, development set, and test set is 18:1:1.

**Table 3.** Sample data in the Service Classification experiment.

| Intro | Tag |
|---|---|
| Let Vimeo take care of all your video needs. Their API handles uploading, transcoding, hosting and playback of any video type. Embed the videos in your own website, or share them with the large community of filmmakers that call Vimeo home. The API uses RESTful calls and responses are formatted in JSON. | Video |
| Let Vimeo take care of all your video needs. Their API handles uploading, transcoding, hosting and playback of any video type. Embed the videos in your own website, or share them with the large community of filmmakers that call Vimeo home. The API uses RESTful calls and responses are formatted in JSON. | Community |
| DataMotion Direct Messaging", "intro": "The DataMotion Direct Messaging API in REST architecture can facilitate methods to send, receive, and manipulate a user's direct messaging inbox. Developers can use a Key to authenticate and make calls that will return JSON formats. DataMotion provides secure and compliant messaging services. | Email |
| DataMotion Direct Messaging", "intro": "The DataMotion Direct Messaging API in REST architecture can facilitate methods to send, receive, and manipulate a user's direct messaging inbox. Developers can use a Key to authenticate and make calls that will return JSON formats. DataMotion provides secure and compliant messaging services. | Compliance |

### 4.3.2. Neural Network Structure

We use a network structure similar to that used in the semantic relationship detection experiment. The difference is that we replaced BERT with AlBERT, which performs better in downstream tasks, in the hope of achieving better results. The network will output the top five probabilities at the same time instead of outputting a single type of label, because a service often belongs to multiple categories. Multiple outputs are more reasonable.

### 4.3.3. Results

When we calculate the accuracy rate, as long as the five tags with the highest probability of the output contain the correct tag, the prediction is considered correct. During the model training process, the accuracy rate changes, as shown in the Figure 9.

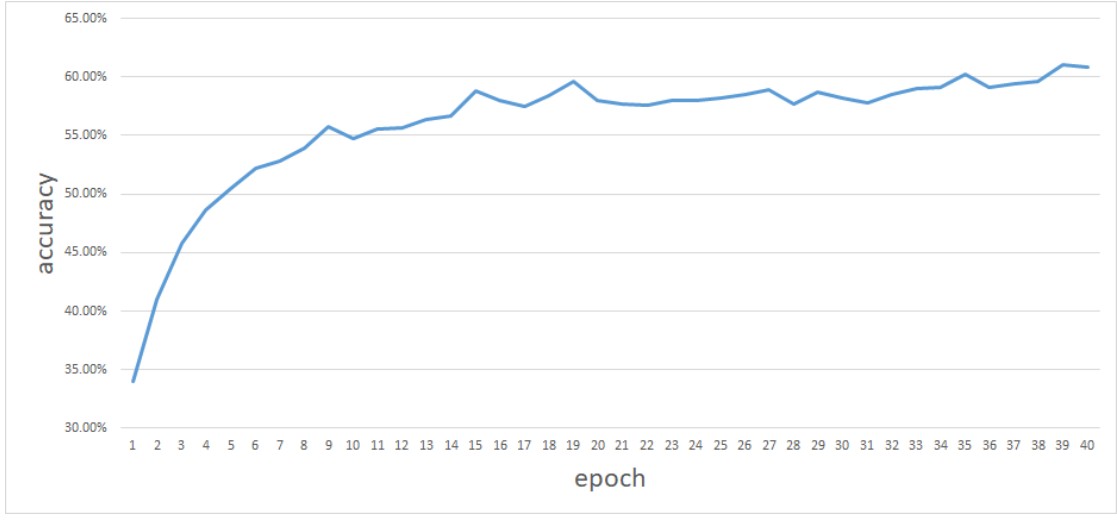

**Figure 9.** The accuracy changes with epochs.

We have achieved a 60.8% accuracy rate on classification tasks with more than 450 categories. This is mainly due to three reasons. One is insufficient training data for training the network. The semantic context information is insufficient. The second is the excessive number of categories. Because there are too many types, it takes a long time for the network to learn the context corresponding to each type, so the accuracy rate improves slowly. The third is the existence of ambiguity. The same context can be mapped into multiple categories. This in itself makes the network confuse the context to a certain extent. This is also confirmed by the decrease in the accuracy of the appearance during the training process.

After getting the network prediction, we pass the output to the translate module and convert it into a graph query language. The accuracy of network prediction is the accuracy of service label generation at the time of service discovery.

### 4.4. Evaluation

This is not the first time that the idea of using knowledge graph to store service information and constructing service ontologies to represent service resources has appeared. Under the guidance of this kind of idea, we propose a brand-new framework, which has stronger semantic extensibility compared with other frameworks of the same purpose. More semantic information about services is stored in the graph. Natural language can be directly used to retrieve the service information stored on the knowledge graph, and the satisfaction degree of retrieval is quantified.

Compared with the framework designed by Guodong et al. [17] We designed a more complex schema layer of the knowledge graph, which can store more detailed information about the service. For example, you can compare the input and output of each method in detail. Service ontology contains only five types of content in the knowledge graph designed by Guodong et al. Service entities are extracted from structured and semi-structured data through knowledge extraction referring to this ontology. The large-scale service knowledge graph we built contains datatype, message, method, among other ontologies, which can help us express more complex information of service and provide us with more options for retrieval. We did not use machine learning to automatically extract entities from the data, but directly convert WSDL documents into service entities. This reduced the uncertainty of the information stored in knowledge graph, so that the constructed service entity will not contain any faults in the process of converting the storage method. To a certain extent, the form of data source is limited to WSDL documents, but the stable conversion can ensure that the accuracy of the final retrieval is only related to the accuracy of the classification. The design uses accuracy to quantify the satisfaction degree of user retrieval.

Compared with the method of the combination of the ELMo representation and CNN proposed by Huang et al. [18], we use the transformer, which has stronger feature extraction capabilities, as the basic unit of pre-training part of network. Moreover, the network extracts features from the natural language profile of service and classifies them to certain semantic tags. We extend the retrieval mechanism from the comparison of the similarity between the profile and the introduction to the classification of arbitrary semantic content. In the experiment of semantic relationship detection, the accuracy rate reached we achieved an accuracy of 95.1%. The highest accuracy rate is 92.9% in semantic textual similarity task on STS-B, which has the same form of input and output compare with semantic relationship detection task. It shows that the classification ability of our network has reached state of the art.

When performing service discovery, Guodong et al. firstly classify the user's intent. They perform service entity recognition after that. Then they construct query templates and calculate the similarity. Our input processing for service discovery is different from the method adopted by Guodong et al. [17]. First, people need to determine the semantic aspect which needs to be the retrieval, and then we map the input natural language profile directly into the tag collection through the network. This is caused by the uncertainty regarding the way in which custom semantic content is expressed. For example, the classification

information of services is expressed in the form of tag attributes, and whether there is an association between services is expressed as whether the relationship named "association" is established or not. In case the user does not know the type of the specific semantic tag, we could also respond to the user's input and judge the category based on the user's description of the service. Through classification of the service description, the translate module will generate a query for retrieval, and finally get the search results. We improve the transparency of the system when solving service discovery tasks.

## 5. Conclusions

The ontology and relationship proposed in the third section are necessary to express the basic information of services. Users can create some special-purpose ontology and relationship by specifying the namespace when constructing the ontology, which can be used to express the specific information of some services. Although different user-defined ontologies have differences, such differences come from the characteristics of services. Because the ontologies and relationships used to represent the basic information of different services are the same, different services can adopt the same pattern when processing the same basic information of different services. The same pattern means a lot of reuse of programs. The system uses knowledge graphs for service representation, stores relevant information about service resources, and has very good scalability. At the same time, since the design of a special service ontology is allowed in the graph, it will also meet the special needs of representation for some of the service resource information.

The scalability of the system is not only reflected in the ability to express traditional service resource information. Compared with the traditional representation of service information using WSDL, the representation of service information on the knowledge graph has better scalability and can store information more conveniently. With the help of neural networks, we can express and use new types of service information, which can contain more semantics. Using additional semantic information, one could perform more fine-grained searches. Compared with the static search conditions in traditional methods, we can define new types of information as search conditions, making service discovery more flexible and more in line with people's needs. When using this information to complete service discovery tasks or service composition tasks, the results will be more in line with people's expectations. Therefore, combining the knowledge graph's ability of expressing service resource information with deep learning technology can make the knowledge graph expandable at the semantic information level. Knowledge graph allows the representation of services to contain more semantic information, and uses this information in subsequent service discovery and composition tasks.

Using deep learning technology, we can bridge the gap between standardized language and natural language, and perform natural language-based service discovery with high accuracy. Deep learning lowers the threshold for service discovery. Various pre-training models can reduce our cost in developing new semantic relations. We do not need to use a very huge database to train the network from the initial state.

When we use the extended semantic information on the knowledge graph for service discovery, there is no uncertainty in the conversion of the classified results, and the accuracy of the classification results is the accuracy of the deep neural network prediction, so the probability that the final result meets the expectation is only related to the accuracy rate of the classification network. The accuracy of the final system is converted to the classification accuracy of the neural network. In the traditional service discovery model, people's satisfaction with the search results cannot be measured concretely, but we use the fusion of knowledge graphs and deep neural networks to visualize the satisfaction of users' search results as the accuracy of the classification network, which allows the system to get timely feedback during development. When using knowledge graphs for service representation and semantic information expansion, developers can make timely evaluations of the system and make further decisions. If you want to improve the accuracy,

you need to improve the accuracy of the classification network, which means more data need to be involved in the training of the network.

We also discovered the shortcomings of this model. Although the transfer learning technology can reduce the burden of training a new classification network, it is necessary to construct a special-purpose dataset whenever a new semantic information is expanded. The data set is classified according to the natural language introduction of the service, and the labelled category of the extended semantic information is used as the label. Further, we need to use this data set to train classification deep neural network. Constructing a data set and training a neural network represent a relatively large cost in terms of both labor and time. Every extension of new semantic information requires a module that converts the output of neural network into the corresponding graph query language. While expanding the number of classification network, it is necessary to construct translation modules to convert the classification results to the graph query language at the same time. Therefore, although the system can integrate various aspects of semantic information, the cost of semantic extension is relatively high. One network can only output semantic information about one certain aspect at a time. Although it can be retrieved based on this aspect of information, common sense tells us that searching with multiple aspects of information tends to narrow the search space and obtain more accurate search results.

We convert the satisfaction of search results into the accuracy of deep neural network classification. Although training data can be used to improve the accuracy of network classification, the amount of data required for each further increase in accuracy will increase significantly after the accuracy is increased to some certain level. This is due to the characteristics of deep neural networks. Therefore, when deep learning technology lowers the threshold of service discovery, it causes an increase in the cost of improving accuracy.

Based on the current progress, the next step mainly includes two aspects. One is to further narrow the gap between natural language and the standardized language used in traditional service discovery, and improve the effect of service discovery. Specifically, we need to obtain more semantics in the natural language input by the user. First, the deep learning technology is used to identify the semantic information contained in an input, and then the semantic classification network is called to identify the aspects of the semantic in the query, and build a more comprehensive query condition. We expect that the classification results of multiple dimensions can be used to filter out more accurate services.

The second aspect is to perform service composition tasks on a graph, integrating a large number of service resources based on the current service representation, and to study the solution of service composition tasks on the knowledge graph. The current service composition task can only manually select services based on the results of service discovery, and combine the input and output of the services in series. We hope to use the basic service information on the knowledge graph and the later expanded semantic information to automatically complete the service composition task and provide a measurement standard for the result of the service composition task.

**Author Contributions:** Conceptualization, D.M. and J.H.; methodology, Q.S.; software, Q.S.; validation, Q.S.; formal analysis, Q.S.; investigation, Q.S. and D.M.; resources, Q.S.; data curation, Q.S.; writing—original draft preparation, Q.S.; writing—review and editing, Q.S.; visualization, Q.S.; supervision, D.M. and J.H.; project administration, J.H.; funding acquisition, D.M. All authors have read and agreed to the published version of the manuscript.

**Funding:** This work was supported by the National Key Research and Development Program of China under Grant No. 2018YFB0203802.

**Data Availability Statement:** Publicly available datasets were analyzed in this study. This data can be found here: http://inpluslab.com/wsdream/, accessed on 24 March 2021. The data used to train the neural network presented in this study are available on request from the corresponding author. The data are not publicly available due to the interests of certain websites.

**Conflicts of Interest:** The authors declare no conflict of interest.

**Abbreviations**

The following abbreviations are used in this manuscript:

| | |
|---|---|
| WSDL | Web Services Description Language |
| RDF | Resource Description Framework |
| BPTT | Backpropagation through Time |
| RNN | Recurrent Neural Network |
| CNN | Convolutional Neural Network |
| BRNN | Bidirectional Recurrent Neural Networks |
| LSTM | Long Short-Term Memory |
| GRU | Gated Recurrent Unit |
| BiLSTM | Bidirectional Long Short-Term Memory |
| BERT | Bidirectional Encoder Representations from Transformers |
| ELMo | Embeddings from Language Models |
| LDA | Latent Dirichlet Allocation |
| SQuAD | Stanford Question Answering Dataset |
| DAML-S | DARPA Agent Markup Language for Services |
| SOAP | Simple Object Access Protocol |
| JSON | JavaScript Object Notation |
| MNLI | Multi-Genre Natural Language Inference |
| QNLI | Question Natural Language Inference |
| MRPC | Microsoft Research Paraphrase Corpus |
| RTE | Recognizing Textual Entailment |
| STS | Semantic Textual Similarity |
| STS-B | Semantic Textual Similarity Benchmark |
| QQP | Quora Question Pairs |

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
