# Peer review of "A Framework for Service Semantic Description Based on Knowledge Graph"

_electronics, doi:10.3390/electronics10091017_

Round 1
Reviewer 1 Report
The paper presents a system which obtains a large-scale knowledge graph for Web Services. The title of the paper is quite general, while the contents of the paper are more specific, so that, the title may be improved to describe the specificity of the work done.
Introduction Section describes what the paper pretends while section 2 addresses the state of art. Both sections are correct, but they seem a bit disorganized, and make it difficult to understand what are the aims of the paper. Besides, state of art section does not compare the approach proposed in the paper with previous works, or even mention if there are such works. It will be very interesting to state this, as a way of making explicit the contributions of the paper in its context.
Sections 3 describes the system and the methods used. The description is quite complete and introduces the information needed for understanding the system. But, it is descriptive, without the technical details that must be expected.
In section 4, the graph obtained by the system and the process followed to build it are shown. The figures of accuracy achieved during the training period and the final rate are also provided in the section.
In my personal opinion, the paper lacks a detailed comparison of the system with similar ones, and also some kind of experts evaluation who can state the interest and capability of the system to obtain the expected results. A complete evaluation section must be provided.
Conclusions resume correctly the results of the study but it fails in introducing the weaknesses and open issues the study presents. The final impression is that the work is still in progress and must be improved for achieving relevant conclusions.
Finally, a revision of writing standard would be necessary. Some passages are difficult to understand. A general review of the language used in the paper by a native speaker may improve the overall legibility.
Reviewer 2 Report
A new Semantic Extension of Knowledge Graph method for web services is proposed in the article. The core of the method is described in the graph (Fig. 4). The article has a good level, I have comments on only two followed chapters:
Related works section:
There is not information provided how the publications for the literature review were selected. It is recommended to add the information concerning the research sampling process for the literature review. It should be made explicit whether literature review is systematic or narrative.
Conclusion section:
It is suggested to describe the implication of the research (‘added value’) for the development of theory and business practice.
Limitations of the study should be revealed and discussed.
Recommendations for further research and proposals of research avenues should be provided.
